# Achieving Noise Robustness by additive normalization of labels

## Abstract

As machine learning models scale, the demand for large volumes of high-quality training data grows, but acquiring clean datasets is costly and time-consuming due to detailed human annotation and noisy data filtering challenges. To address this, symmetric loss functions were introduced in the context of label noise, enabling models trained on noisy data to perform comparably to those trained on clean data without explicit noise knowledge. Loss functions satisfying a specific symmetry condition exhibit robustness to label noise. Building on this, we propose a novel method to derive noise-robust loss functions using monotonic functions and label normalisation, which involves a simple normalisation of labels that leads to noise robustness when labels are corrupted. Unlike other approaches, this method allows creation of new loss functions by defining application-specific monotonic functions rather than relying on predefined losses. We formally prove their theoretical properties, propose two concrete noise-robust losses, and demonstrate through extensive empirical evaluations on computer vision and natural language processing tasks that our losses outperform standard and existing noise-robust losses. Our evaluations indicate better learning of *decision boundaries*, *faster convergence*, and *improved robustness to noise* using the proposed loss functions.

## 1 Introduction

The scale of machine learning models has grown dramatically in recent years, fueling remarkable advances across diverse applications such as computer vision, natural language processing, and speech recognition. This rapid growth is primarily due to the advent of transformers (Vaswani et al., 2017), which are trained to predict the next token in a sequence. These transformer models are data-hungry and require vast amounts of corpus data for effective training.

High-quality datasets (Sajith et al., 2024) are essential for training these models effectively, but obtaining such data is often prohibitively expensive and time-consuming. This is mainly because labeling large datasets demands meticulous human annotation and expert domain knowledge, especially for complex tasks. Even in unsupervised learning settings, corpora tend to be noisy due to human errors or corruption during data acquisition from various sources. Moreover, real-world datasets frequently contain noisy or incorrect labels caused by human errors, ambiguity in labeling criteria, or automated labeling processes. Label noise poses a significant challenge because it can cause models to overfit incorrect information, which degrades predictive performance and harms generalization. The problem of label noise is particularly critical in fine-tuning, where models are trained on smaller, domain-specific datasets, making them more sensitive to imperfect labels.

Deep neural networks are particularly susceptible to label noise given their large capacity to memorize training examples, including mislabeled ones. This phenomenon can lead to performance deterioration even when only a fraction of labels are corrupted. Prior work has explored numerous approaches to mitigate the impact of label noise, including data cleaning (Bernhardt et al., 2022), data filtering (Wu et al., 2020), re-weighting samples, and architectural innovations (Jindal et al., 2019; Li et al., 2020; Vashisht et al., 2024; Chen et al., 2020). Among these strategies, the design of noise-robust loss functions has emerged as a theoretically grounded and practically effective direction. Loss functions play a critical role in shaping the learning process by quantifying the discrepancy between predictions and labels, and certain loss functions inherently exhibit robustness to noise by reducing the influence of incorrect labels.

A notable theoretical advancement in this domain was introduced by Ghosh et al. (2017), who formalized a symmetry condition that loss functions should satisfy to guarantee robustness to label noise. Their framework demonstrated that models trained with such symmetric loss functions on noisy data could achieve error rates comparable to models trained on clean datasets, without requiring explicit knowledge or estimation of the noise characteristics or datasets. This work has lead to many follow up investigations. One of these investigations is the concept of normalizing (Ma et al., 2020b) loss functions. In this work, any loss function can be divided by sum of losses for all class-labels to obtain a noise robust loss function. But, the authors reported that this formulation lead to underfitting Ma et al. (2020b), and hence had to be added with an active loss (possibly non-noise robust) to make the training faster. A similar work Paquin et al. (2024) does the same, but not dividing but by subtraction of losses summed over all labels. But, these methods only give conditions for obtaining losses from existing losses. This insight has informed the construction of novel noise-robust loss functions that offer resilience to common types of label corruption.

Building on this foundation, our work introduces a novel methodology for deriving noise-robust loss functions using (only) monotonic functions. We formalize the theoretical properties of these transformations, proving that they preserve the symmetry condition necessary for noise robustness. Further, we present two specific instantiations of noise-robust losses derived from this framework, tailored to balance robustness and optimization tractability. To validate our approach, we conduct extensive empirical experiments spanning computer vision and natural language processing benchmarks. Our proposed losses consistently outperform both classical baseline losses and recent state-of-the-art noise-robust losses in the literature, demonstrating superior robustness and improved model accuracy under various noise regimes.

This study contributes a principled yet practical approach to mitigating label noise effects, enabling reliable training of large-scale models in noisy real-world settings. By integrating theoretical rigor with empirical validation, our proposed framework advances the understanding and application of noise-robust learning in modern machine learning. Our contributions can be summarized as follows:

- Development of a framework that allows us to design novel application-specific noise-robust loss functions (Section 4)
- Two novel loss functions designed by the above strategy, along with rigorous experimental evaluation. (Section 4.2)
- Our evaluations indicate that the designed loss functions outperform other loss functions in various tasks and show a lesser degradation in performance with an increase in noise level in datasets. Moreover, models trained using our loss functions learn better decision boundaries (Figure 1 and Figure 3) and converge faster compared to other methods (Figure 2).

## 2 PRIOR WORK

The quality of a trained machine learning model is primarily influenced by the optimization landscape shaped by the chosen loss function during training. Loss functions guide the model toward achieving specific objectives aligned with the task. For example, Intersection over Union (IoU) loss Rezatofighi et al. (2019) caters to object detection, cross-entropy loss to classification, and Kullback–Leibler (KL) divergence to distribution matching.

Among various loss functions, some have been shown to be robust against label noise Ghosh et al. (2017), enabling models to learn meaningful patterns despite corrupted labels. However, many such noise-robust loss functions are independently designed and are not easily adaptable to existing losses.

To expand applicability, two important approaches have been proposed to convert existing loss functions into noise-robust variants: normalization and symmetrization.

**Normalization**, (Ma et al., 2020b) in applies a simple normalization step to any loss function to theoretically guarantee robustness to noisy labels. Their work proves that by normalizing losses so that their values sum to a constant over classes, all losses achieve noise tolerance. However, practical use revealed an underfitting problem where normalized robust losses suffered reduced accuracy due to diminished learning ability. To mitigate this, they proposed the Active Passive Loss (APL) framework combining two robust loss functions that mutually improve training effectiveness. Despite advancing robustness theory and empirical performance, normalization-based methods still depend on existing losses as starting points and can underfit if not carefully combined.

**Symmetrization**, studied in (Paquin et al., 2024), constructs noise-robust losses by making them symmetric with respect to class labels. Symmetric losses treat all misclassification errors uniformly, thereby diminishing the impact of label noise. This approach often involves averaging a loss function with its complement or designing inherently symmetric losses. While effective, symmetrization techniques require a base loss function with proper structural properties to be converted, thus limiting their applicability only to certain (permutation invariant) losses.

Despite these advances, the need for a pre-existing base loss function to convert into noise-robust forms constrains the generality of these methods. Furthermore, applicability can be limited in settings like reinforcement learning, where the concept of a classical loss function is ambiguous. Reinforcement learning focuses on maximizing cumulative rewards rather than minimizing explicit losses derived from ground truth labels, making direct application of normalization and symmetrization techniques challenging.

## 3 PRELIMINARIES

Let $\mathcal{X} \subseteq \mathbb{R}^d$ denote the feature space from which the examples are drawn, and let $\mathcal{Y} = \{\mathbf{e}_1, \ldots, \mathbf{e}_k\}$ be the set of one-hot encoded class labels, where each $\mathbf{e}_i \in \{0,1\}^k$ is a vector with 1 in the $i$-th position and 0 elsewhere.

A classifier learning problem is defined by training data

$$S = \{(x_1, \mathbf{y}_{x_1}), \ldots, (x_N, \mathbf{y}_{x_N})\} \subseteq (\mathcal{X} \times \mathcal{Y})^N,$$

drawn i.i.d. from an unknown distribution $D$ over $\mathcal{X} \times \mathcal{Y}$.

We represent a classifier as $h(x) = \text{pred} \circ p(x)$, where $p : \mathcal{X} \to \mathcal{C}$, $\mathcal{C} \subseteq \mathbb{R}^k$, and the function $\text{pred} : \mathcal{C} \to \mathcal{Y}$ predicts the class label from $p(x)$. For simplicity, we refer to $p$ itself as the classifier.

A loss function is a map $\ell : \mathcal{C} \times \mathcal{Y} \to \mathbb{R}^+$.

**Clean risk** is the expected loss evaluated under the true, noise-free distribution $D$:

$$R_\ell(p) = \mathbb{E}_{(x,\mathbf{y}) \sim D}\big[\ell(p(x), \mathbf{y}_x)\big].$$

In the presence of label noise, the available data is noisy,

$$S_q = \{(x_n, \tilde{\mathbf{y}}_{x_n}), n = 1, \ldots, N\},$$

where the noisy label $\tilde{\mathbf{y}}_x$ satisfies

$$\tilde{\mathbf{y}}_x = \begin{cases} \mathbf{y}_x, & \text{with probability } 1 - q_x, \\ \mathbf{e}_j \neq \mathbf{y}_x, & \text{with probability } \bar{q}_{xj}, \end{cases}$$

with noise rates $q_x$ and $\bar{q}_{xj}$ such that $\sum_{j \neq i} \bar{q}_{xj} = q_x$ for $\mathbf{y}_x = \mathbf{e}_i$.

The **noisy risk** is the expected loss evaluated on the noisy label distribution $D_q$,

$$R_\ell^q(p) = \mathbb{E}_{(x,\tilde{\mathbf{y}}) \sim D_q}\big[\ell(p(x), \tilde{\mathbf{y}}_x)\big].$$

Let $p^*$ and $p_q^*$ be minimizers of the clean risk $R_\ell(p)$ and noisy risk $R_\ell^q(p)$, respectively.

Risk minimization under loss $\ell$ is said to be noise-tolerant if

$$\Pr_D\big[\text{pred} \circ p^*(x) = \mathbf{y}_x\big] = \Pr_D\big[\text{pred} \circ p_q^*(x) = \mathbf{y}_x\big].$$

## 4 PROPOSED METHOD: ADDITIVE NORMALIZATION OF LABELS

We formulate loss functions for a $k$-class classification problem. Consider the loss function as

$$\ell(\mathbf{y}, (\mathbf{p})) = -\langle \mathbf{y}, \mathbf{f}(\mathbf{p}) \rangle.$$

Here, $\mathbf{y} \in \{0,1\}^k$ is the label represented as a one-hot vector, and

$$\mathbf{f}(\mathbf{p}) = \big(f_1(p_1), \ldots, f_k(p_k)\big),$$

where the functions $f_1, \ldots, f_k : \mathbb{R} \to \mathbb{R}$ are strictly increasing monotonic functions, and $p_1, \ldots, p_k$ are model prediction probabilities for each label.

It can be easily seen that minimizing this loss recovers the true label for any choice of monotonic $f_i$, i.e.,

$$\arg\min_{\mathbf{p}} \ell(\mathbf{e}_i, \mathbf{p}) \implies p_i = 1$$

for any one-hot vector $\mathbf{e}_i$, $i \in \{1, \ldots, k\}$.

Examples of losses in this category include cross-entropy loss with $f_i(p) = \log(p)$ and linear loss with $f_i(p) = p$.

Within this structure of losses, we now describe the additive normalization of labels.

## 4.1 ADDITIVE NORMALIZATION OF LABELS

To illustrate the effect of label normalization in the presence of label noise, we begin with a simple binary classification setting. Unlike standard one-hot labels, here the label vectors are defined as transformed one-hot vectors:

$$\bar{\mathbf{y}} \in \{(-1, 1), (1, -1)\}.$$

Noisy labels $\tilde{\bar{\mathbf{y}}}$ are generated by flipping the true label $\bar{\mathbf{y}}$ with probability $q$, uniformly, i.e.,

$$\tilde{\bar{\mathbf{y}}} = \begin{cases} \bar{\mathbf{y}} & \text{with probability } 1 - q, \\ -\bar{\mathbf{y}} & \text{with probability } q. \end{cases}$$

We define the loss function for prediction $\mathbf{f}(\mathbf{p}) = [f_1(\mathbf{p}), \ldots, f_k(\mathbf{p})]$ as the negative inner product of the noisy label and the prediction:

$$\ell(\tilde{\bar{\mathbf{y}}}, \mathbf{f}(\mathbf{p})) = -\langle \tilde{\bar{\mathbf{y}}}, \mathbf{f}(\mathbf{p}) \rangle.$$

Taking the expectation of this loss over the noisy labels conditioned on the true label $\bar{\mathbf{y}}$ yields:

$$\begin{aligned} \mathbb{E}_{\tilde{\bar{\mathbf{y}}}|\bar{\mathbf{y}}} \left[ \ell(\tilde{\bar{\mathbf{y}}}, \mathbf{f}(\mathbf{p})) \right] &= (1 - q)(-\langle \bar{\mathbf{y}}, \mathbf{f}(\mathbf{p}) \rangle) + q(-\langle -\bar{\mathbf{y}}, \mathbf{f}(\mathbf{p}) \rangle) \\ &= (1 - q)(-\langle \bar{\mathbf{y}}, \mathbf{f}(\mathbf{p}) \rangle) + q\langle \bar{\mathbf{y}}, \mathbf{f}(\mathbf{p}) \rangle \\ &= (1 - 2q) \cdot (-\langle \bar{\mathbf{y}}, \mathbf{f}(\mathbf{p}) \rangle) . \end{aligned}$$

This result implies that

$$\mathbb{E}_{\mathcal{X} \times \tilde{\mathcal{Y}}} \ell(\bar{\mathbf{y}}, \mathbf{p}) = (1 - 2q) \mathbb{E}_{\mathcal{X} \times \mathcal{Y}} \ell(\bar{\mathbf{y}}, \mathbf{p}).$$

Hence, when $q < \frac{1}{2}$, the minimizer of the expected *clean* risk coincides with the minimizer of the expected *noisy* risk, demonstrating robustness to noise under this condition.

This analysis generalizes to multi-class classification with $k$ classes as follows. Given a one-hot encoded label $\mathbf{y} \in \{0, 1\}^k$ and a noisy label $\tilde{\mathbf{y}}$ flipped uniformly with probability $q$, define the modified vector label vector by applying **additive normalization of labels** as:

$$\bar{\mathbf{y}} = \frac{1}{k - 1}(k\mathbf{y} - \mathbf{1}), \tag{1}$$

where $\mathbf{1}$ is the all-ones vector in $\mathbb{R}^k$.

The conditional expectation of the noisy transformed label is:

$$\begin{aligned} \mathbb{E}[\tilde{\bar{\mathbf{y}}} \mid \mathbf{y}] &= \frac{1}{k - 1} \left( k \, \mathbb{E}[\tilde{\mathbf{y}} \mid \mathbf{y}] - \mathbf{1} \right) \\ &= \frac{1}{k - 1} \left( k \left( (1 - q)\mathbf{y} + \frac{q}{k - 1}(\mathbf{1} - \mathbf{y}) \right) - \mathbf{1} \right) \\ &= \frac{1}{k - 1} \left( \left( k - \frac{k^2 q}{k - 1} \right) \mathbf{y} + \left( \frac{kq - k + 1}{k - 1} \right) \mathbf{1} \right) \\ &= \left( 1 - \frac{kq}{k - 1} \right) \bar{\mathbf{y}}. \end{aligned}$$

Accordingly, the expected loss with prediction $\mathbf{f}(\mathbf{p})$ is:

$$-\langle \mathbb{E}[\tilde{\bar{\mathbf{y}}} \mid \mathbf{y}], \mathbf{f}(\mathbf{p}) \rangle = -\left( 1 - \frac{kq}{k - 1} \right) \langle \bar{\mathbf{y}}, \mathbf{f}(\mathbf{p}) \rangle . \tag{2}$$

This results in the following theorem:

**Theorem 1.** *Consider any loss function defined as*

$$\ell(\mathbf{y}, \mathbf{p}) := -\langle \bar{\mathbf{y}}, \mathbf{f}(\mathbf{p}) \rangle,$$

*where $\bar{\mathbf{y}}$ is defined by the additive normalization in equation 1. Then, the minimizer of the* clean *risk is the same as the minimizer of the* noisy *risk provided the noise probability satisfies*

$$q < \frac{k-1}{k}.$$

*Proof.* The proof directly follows from eq. (2) □

**Some remarks on the above theorem** 1. It can be verified that even with the proposed normalization minimization of the losses recovers the true labels due to the monotonic increasing property of $f_i$s. 2. The label normalization leads to the condition that $\sum_{\mathbf{y} \in \mathcal{Y}} \ell(\mathbf{y}, \mathbf{p}) = 0$. Therefore by the robustness theorem by Ghosh et al. (2017) the proposed normalization is also robust to *class conditional noise* and *instance dependent noise*.

### 4.2 Loss functions designed by the above normalization

**Noise Robust Focal Loss (NRFL):** NRFL is a noise-robust adaptation of the popular focal loss. The focal loss is defined as $L_{\text{FL}}(p) = -(1-p)^\gamma \log p$, which down-weights well-classified examples to focus training on hard samples. Incorporating focal loss into our noise robustness framework, the noise-robust version (NRFL) is computed as

$$\ell(p, e_t) = -L_{\text{FL}}(p_t) + \sum_{\substack{j=1 \\ j \neq t}}^{k} \frac{1}{k-1} L_{\text{FL}}(p_j),$$

where $p_t$ is the predicted probability for the true class $t$, and the summation averages over the probabilities of other classes. This formulation balances loss contributions to improve robustness against label noise.

**Weighted Robust Log Loss (WRLL):** WRLL is derived from Robust Log Loss designed to solved the class imbalance problem. This is done by computing the weights $\alpha$ value for each class as $\alpha_i = \frac{1}{frequency\ of\ i^{th}\ token}$. According to our framework, the label-specific loss is computed as $f_i(p) = \log(\alpha_i + p)$. Thus, WRLL is computed as

$$L(p, e_t) = -f_i(p_t) + \sum_{j=1, j \neq t}^{k} \frac{1}{k-1} f_i(p_j)$$

The main advantage of WRLL over RLL is that the loss computed for the $i^{th}$ label is dependent on its frequency in the training data.

**Advantages of Label Normalization over Loss Normalization**

- Label normalization can be applied in scenarios such as reinforcement learning, where a well-defined loss function may not exist but labels or target signals remain well-defined and meaningful.
- It is computationally simpler to implement, as it involves straightforward transformations on labels, whereas loss normalization often requires additional complex computations during model training.

## 5 Experiments

This section presents experimental results comparing NRFL and WRLL against standard and noise-robust loss functions including Cross Entropy (CE), Mean Absolute Error (MAE), RLL, and Normalised Focal Loss (NFL) (Ma et al., 2020a). Extensive experimentation has been conducted across two major domains: computer vision and natural language processing. For datasets without a predefined validation set, we partition the original training set into training and validation subsets using an 80:20 split. Details of the experimental setup, including hyperparameters and the methodology for introducing noise into different datasets, are provided in Appendix A.

In all result tables, we highlight the top two performing methods: the best-performing method is shown in **bold**, while the second-best is indicated with a grey highlight . Next, we present our evaluation on object classification tasks.

### 5.1 Evaluation on Object Classification Tasks

To assess the robustness of NRFL and WRLL in the presence of label noise, we train separate models using these loss functions, along with the baselines described in Section 4, on three benchmark datasets: Modified National Institute of Standards and Technology (MNIST) (LeCun et al., 1998), Fashion-MNIST (Xiao et al., 2017), and Canadian Institute for Advanced Research-10 (CIFAR-10) (Krizhevsky, 2009). Each dataset consists of approximately 60k training samples, 10k testing samples, and 10 classes. For every dataset, we introduce label noise at two levels (30% and 60%), and create two instances of each noise level. We report both the mean accuracy and its standard deviation across 2 instances of each noise variant.

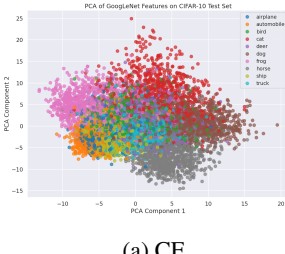 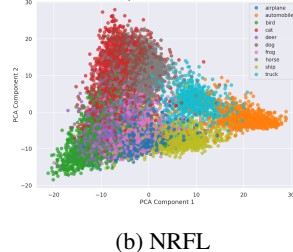 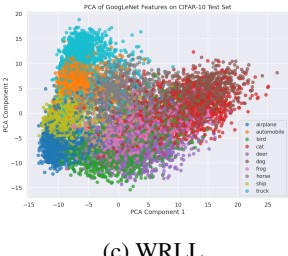

(a) CE        (b) NRFL        (c) WRLL

Figure 1: Comparison of decision boundaries for GoogleNet trained on 50% noisy CIFAR-10 using: (a) CE (b) NRFL (c) WRLL

In our initial set of experiments (Table 1), we trained GoogleNet for 15 epochs on each of the three datasets under varying noise conditions. Each experiment was repeated twice on each noise level and the mean and standard deviation of the accuracy are reported in Table 1. We observe that models trained with noise-robust loss functions demonstrate greater resilience to noise, maintaining similar or only slightly degraded performance with an increase in noise level. This trend holds across all methods in Table 1, with the exception of the standard cross-entropy loss, which is not a noise-robust training technique. Among the compared loss functions, NRFL and WRLL consistently achieve better performance in most evaluation scenarios.

| Loss Type | MNIST | | | CIFAR10 | | | Fashion MNIST | | |
|---|---|---|---|---|---|---|---|---|---|
| | 0% | 30% | 60% | 0% | 30% | 60% | 0% | 30% | 60% |
| CE | *99.47* | 98.71 | 97.76 | *82.67* | 76.1 | 73.41 | 92.63 | 90.97 | 86.47 |
| | | (5.30) | (1.81) | | (3.48) | (4.29) | | (0.35) | (5.93) |
| MAE | 99.33 | 98.74 | **99.26** | 81.44 | *79.28* | *79.11* | 90.26 | **92.22** | **92.13** |
| | | (0.08) | **(0.06)** | | (1.58) | (0.52) | | **(0.61)** | **(0.25)** |
| NFL | 99.4 | 95.32 | 93.55 | 80.49 | 74.56 | 68.83 | 92.86 | 90.61 | 86.84 |
| | | (4.29) | (7.01) | | (2.65) | (3.13) | | (0.43) | (4.48) |
| RLL | 99.34 | *99.11* | **99.26** | 82.35 | 78.6 | 73.93 | 93.09 | 89.83 | *91.79* |
| | | (0.2) | **(0.18)** | | (4.58) | (6.75) | | (3.75) | (2.6) |
| GCE | 98.27 | 98.42 | 90.14 | 76.82 | 66.99 | 46.44 | 90.43 | 87.42 | 84.79 |
| | | (0.7) | (0.12) | | (0.32) | (0.91) | | (0.26) | (0.86) |
| APL | 99.35 | 98.23 | 98.18 | 71.19 | 78.27 | 50.53 | 87.05 | 89.54 | 88.84 |
| | | (0.84) | (0.12) | | (0.41) | (0.96) | | (0.72) | (0.09) |
| WRLL* | **99.59** | 98.78 | *99.12* | 79.93 | **79.49** | 77.93 | **93.47** | *91.62* | 90.12 |
| | | (0.66) | (0.22) | | **(4.86)** | (5.76) | | (1.76) | (4.23) |
| NRFL* | 99.38 | **99.21** | 98.82 | **82.71** | 78.83 | **79.66** | *93.28* | 89.46 | 81.87 |
| | | **(0.28)** | (0.5) | | (3.6) | **(3.6)** | | (3.52) | (2.68) |

Table 1: Accuracy of GoogleNet trained on various datasets after 15 epochs

To determine the reason for the superior performance of NRFL and WRLL, we analysed the representations learned by GoogleNet (Szegedy et al., 2015) trained on the CIFAR-10 dataset with 60% label noise, using CE, NRFL, and WRLL loss functions. Feature representations were extracted from the trained models on the CIFAR-10 test set, and Principal Component Analysis (PCA) was applied to project them into two dimensions for visualisation (Figure 1). As shown in Figure 1,

models trained with NRFL and WRLL exhibit clearer class separation compared to CE, forming distinct clusters aligned with CIFAR-10 labels. Beyond qualitative inspection, we quantified clustering quality using the Silhouette Score (Rousseeuw, 1987), obtaining values of 0.046 for CE, 0.18 for NRFL, and 0.1539 for WRLL. These results confirm that NRFL and WRLL yield more well-defined decision boundaries than CE even under severe label noise (60%).

| Loss Type | MNIST | | | CIFAR10 | | | Fashion MNIST | | |
|---|---|---|---|---|---|---|---|---|---|
| | 0% | 30% | 60% | 0% | 30% | 60% | 0% | 30% | 60% |
| CE | 73.5 | 61.35 | 58.49 | 34.43 | 28.83 | *27.39* | 73.3 | 67.2 | 65.54 |
| MAE | 53.81 | 56 | 35.23 | 23.68 | 22.78 | 15.8 | 69.37 | 61.41 | 53.39 |
| RLL | 75.51 | 72.05 | 62.9 | 33.55 | 31.6 | 25.83 | 74.45 | *71.95* | *68.1* |
| NFL | 65.27 | 46.37 | 42.56 | 32.03 | 28.84 | 23.05 | 72.2 | 68.56 | 64.03 |
| NRFL | *80.29* | *70.98* | *67.03* | *34.61* | *31.85* | 26.61 | *75.63* | 68.83 | 67.03 |
| WRLL | **81.03** | **78.46** | **68.92** | **37.57** | **33** | **32.4** | **79.59** | **72.79** | **70.73** |

Table 2: Accuracy of Early Stopping of ResNet trained on various datasets after 30 epochs

In our second set of experiments, we employed a larger architecture than GoogleNet, namely ResNet18 He et al. (2016), which consists of around 11.7 million parameters. Following a methodology similar to that of GoogleNet, ResNet18 was trained for 30 epochs with early stopping. The hyperparameter $\gamma$ for NRFL was fixed at 0.01. Owing to the higher computational cost of training ResNet, only a single instance per noise level was used. As reported in Table 2, models trained with WRLL and NRFL consistently outperform other methods across most scenarios. This superior performance can again be attributed to the more robust decision boundaries learned with these losses, as reflected in Figure 1 and the corresponding Silhouette scores. We have also conducted similar experiments for ResNet-18 on the CIFAR-100 (Krizhevsky, 2009) dataset, which are presented in Appendix B.2. In addition to improved decision boundaries, models trained with the proposed

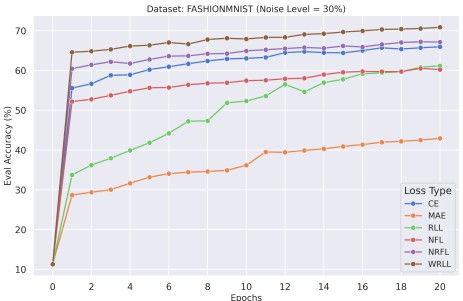

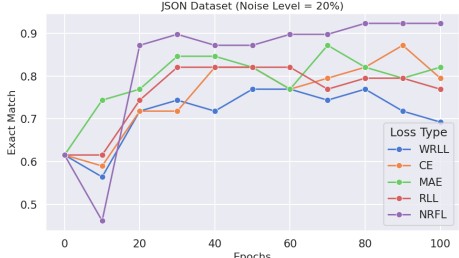

(a) ResNet on Fashion MNIST (20% noise)      (b) LLaMa on JSON dataset (20% noise)

Figure 2: Convergence trajectory of different models trained using different loss functions

NRFL and WRLL loss functions demonstrate faster convergence compared to alternative methods. This behaviour is illustrated in Figure 2a, which shows that across training epochs, models trained with WRLL and NRFL consistently outperform other loss functions. These results suggest that models trained using NRFL and WRLL learn decision boundaries much faster than other methods. Following this analysis, we extend our evaluation to natural language processing (NLP) tasks to assess the generalizability of our findings.

## 5.2 EVALUATION ON NLP TASKS

We evaluate the proposed methods on five natural language processing (NLP) tasks arranged in increasing order of difficulty. Experiments are conducted with two distinct models: `meta-llama/Llama-3.2-1B-Instruct` (Meta AI, 2024) and `Qwen/Qwen2.5-0.5B-Instruct` (Team, 2025). In the subsequent text, we refer to these models as LLaMA and Qwen, respectively.

**NLP-based classification task (News20 Dataset) :** Extending on our computer vision experiments, we next explore whether similar trends hold for NLP classification tasks with Large Language Models (LLMs). To do so, LLaMA was trained on the News20 dataset (Lang, 1995) with 20% label

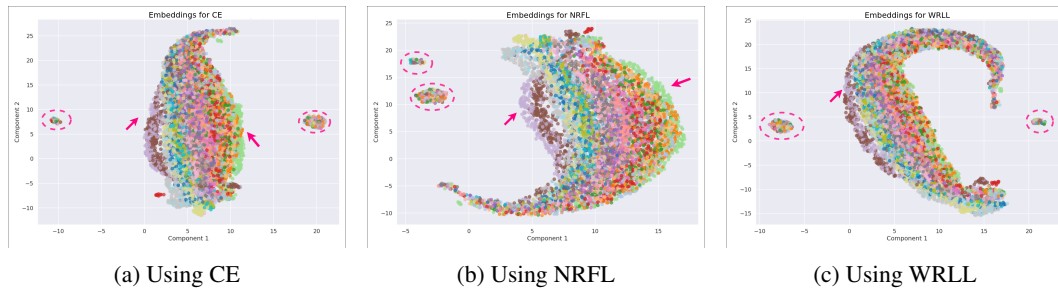

| (a) Using CE | (b) Using NRFL | (c) Using WRLL |

Figure 3: Decision boundaries learnt by LLaMA on 20% noisy News20 dataset

| Loss Type | Accuracy | Calsinki Score |
|-----------|----------|----------------|
| CE | 45.46 | 38.67 |
| NRFL | **63.95** | **39.05** |
| WRLL | 56.72 | 20.91 |

Table 3: LLaMa trained on 20% noisy News20 dataset

noise. Table 3 reports the metrics for models trained with CE, NRFL, and WRLL. Among these, NRFL attains the highest accuracy and the best Calinski score (Caliński & Harabasz, 1974), indicating that it learns the best representations among all three. Figure 3 presents a visual representation of the embeddings learned by LLaMA. This is consistent with Table 3 as models trained with NRFL outperform those trained with CE and WRLL. As shown in the figure, the light violet and brown clusters, as well as the light green and orange clusters, are more clearly separated under NRFL. In particular, the small cluster of points marked in dashed pink circle is far more separated in NRFL than other methods, indicating that NRFL enables the model to separate points belonging to different classes. Both this and the computer vision experiments establish the fact that loss functions derived from our framework improve classification performance. Next, we move on to a more challenging task, which is the information extraction task in NLP.

| Loss Type | **LLaMa** | | | **Qwen** | | |
|-----------|------|------|------|------|------|------|
| | 0% | 20% | 50% | 0% | 20% | 50% |
| CE | *84.62* | *82.48 (1.81)* | *80.77 (5.44)* | 33.33 | 30.77(1.81) | 26.92(5.44) |
| MAE | *84.62* | 79.49 (10.88) | 79.49 (7.26) | **79.49** | **79.49(3.63)** | **79.49(3.63)** |
| RLL | 82.05 | 78.20 (5.44) | 78.20(9.06) | *76.92* | *75.64(1.81)* | *74.36(7.25)* |
| NFL | 51.28 | 51.28 (0.0) | 46.15 (9.07) | 38.46 | 34.62(5.44) | 28.21(5.44) |
| GCE | 21.8 | 20.7 (1.21) | 20.7 (2.01) | 15.3 | 13.5 (1.2) | 12.8 (1.5) |
| APL | 17.2 | 13.8 (1.32) | 10.3 (2.14) | 11.2 | 10.1 (1.4) | 9.6 (1.4)) |
| NRFL | **92.31** | **92.31 (7.25)** | **84.62 (1.81)** | 71.79 | 69.23(3.63) | 70.51(1.81) |
| WRLL | 79.48 | 76.92 (0.0) | 61.54 (0.0) | 64.10 | 62.67(1.63) | 60.26(5.44) |

Table 4: Accuracy after LLaMa and Qwen are trained on the JSON dataset

**Information Extraction Task (Synthetic JSON Dataset):** The information extraction task evaluates a model's natural language understanding capabilities. The LLM must understand a given passage and extract specific entities, such as project names, company names, and person names, which are to be returned as a JavaScript Object Notation (JSON) object. We constructed a synthetic dataset following the methodology described by Shadi Copty[1] . The dataset comprises 144 samples, split into training, validation, and test sets using an 80:10:10 ratio. Table 4 reports the accuracy of LLaMA and Qwen trained on the JSON dataset using various loss functions. For LLaMA, NRFL outperforms the other methods, while for Qwen, MAE and WRLL achieve higher scores. The better performance of MAE in this case is likely due to the small dataset size, which prevents underfitting and allows MAE to perform well. Figure 2b displays the convergence trajectories of models trained

---

[1]https://huggingface.co/shadicopty/llama3.2-entity

with different loss functions on the JSON dataset with 20% noise. NRFL exhibits stable and efficient convergence, consistent with our observations in the computer vision experiments (Figure 2a). With these observations, we proceed to the next challenging task which is the translation task.

| Loss Type | LLaMa | | | Qwen | | |
|---|---|---|---|---|---|---|
| | 0% | 20% | 50% | 0% | 20% | 50% |
| CE | 51.26 | 51.63 (0.14) | 50.71(0.33) | 39.72 | 32.96 (2.12) | 30.16 (1.12) |
| MAE | **60.37** | **60.32 (0.39)** | **60.80(0.26)** | **62.03** | **61.88 (0.02)** | *60.04 (0.11)* |
| RLL | 57.78 | 57.78 (0.83) | *58.56(0.71)* | 60.40 | 59.27 (0.10) | 58.76 (0.08) |
| NFL | 41.96 | 41.51 (0.60) | 42.01(1.10) | 34.36 | 34.71 (0.82) | 33.38 (0.23) |
| NRFL | *58.64* | *58.75 (0.77)* | 58.27(0.61) | 50.37 | 60.02 (0.53) | 59.22 (0.10) |
| WRLL | 51.42 | 56.35 (7.06) | 49.80(2.78) | *61.72* | *61.77* (0.11) | **61.97 (0.47)** |

Table 5: Logical equivalence score of LLaMa and Qwen trained on MALLS dataset (300 instances)

**Translation Task (MALLS dataset)** The main objective of this task is to convert a natural language sentence into its corresponding first-order logic (FOL) expression. Performance on this task depends on the model's natural language understanding, as it has to first understand the input text and then represent it accurately in FOL form. We use the MALLS dataset Yang et al. (2024) et al. and introduce noise following the methodology described in their work. Following the original study, we report the logical equivalence score, which measures the overlap between the predicted and ground-truth FOL expressions. Table 5 presents these results. For LLaMA, NRFL performs best, while for Qwen, WRLL ranks just after MAE. The better performance of MAE is likely due to the small dataset size, as only 300 instances are considered, limiting underfitting. These observations are consistent with those seen on the JSON dataset. Next we move on to a reasoning task.

| Loss Type | LLaMa | | |
|---|---|---|---|
| | 0% | 20% | 50% |
| CE | **49.2** | 48.8 | 48.8 |
| MAE | *49.1* | 48.7 | 48.7 |
| RLL | 49.0 | 48.5 | *49.1* |
| NFL | 48.9 | **49.2** | 48.9 |
| NRFL | 48.5 | *49.0* | 48.7 |
| WRLL | *49.1* | **49.2** | 49.3 |

(a) OpenBookQA dataset (Accuracy±1.6)

| Loss Type | LLaMa | | |
|---|---|---|---|
| | 0% | 20% | 50% |
| CE | 38.51 | 38.82 | 38.66 |
| MAE | *40.25* | *40.1* | *40.1* |
| RLL | **41.09** | 38.51 | 37.3 |
| NFL | 36.69 | 39.2 | 37.75 |
| NRFL | 39.57 | **40.48** | **40.25** |
| WRLL | 38.28 | 39.27 | 38.36 |

(b) GSM8k Dataset (Accuracy±1.3)

Table 6: Accuracy of LLaMa trained on different datasets on diferent loss functions

**Reasoning Task (GSM8k dataset):** Reasoning is a challenging task for LLMs, as it requires cognitive abilities typically present in living beings such as humans. For this task, we use the GSM8k dataset Cobbe et al. (2021), where the model has to solve a math problem by first providing the reasoning steps and then the final numerical answer in a specified format. Note that we introduce noise only to the final answer by randomly flipping its digits, leaving the reasoning steps intact. While NRFL generally outperforms other loss functions, the performance gap is smaller in this case, likely due to the presence of noise only in the final answer.

**Question Answering (QnA) Task (Openbook dataset):** Question answering requires an LLM to combine natural language understanding with reasoning. The OpenBookQA dataset (Mihaylov et al., 2018) contains questions from various domains ranging from Science, Technology, Engineering and Mathematics (STEM) to general knowledge. Thus, the LLM needs to rely on its prior knowledge, use it to understand the question and then generate the correct answer. Models are trained with early stopping after 10 epochs, and the resulting accuracy scores are reported in Table 6a. As observed, NRFL and WRLL perform well across most cases. We conjecture that training for additional epochs could further increase the performance differences among the methods. We even test on a more challenging problem, which is the automatic short answer grading problem, whose results have been added in Appendix B.1.

## 6 CONCLUSION

Our proposed framework for noise-robust loss functions offers a flexible and theoretically sound approach that improves model performance in noisy label settings. By enabling the design of application-specific losses through monotonic functions and label normalization, this method advances the state of the art in noise robustness, as validated by strong empirical results across diverse tasks.

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

## APPENDIX

## A EXPERIMENTAL SETUP

The hyperparameters used for each and every experiment is mentioned in each subsection below.

### A.1 OBJECT CLASSIFICATION TASK

A learning rate of $10^{-4}$ was used for all loss functions, except for NRFL and WRLL, where a higher learning rate of $10^{-3}$ was adopted, with a batch size of 2048. For NRFL, the gradient update is computed as

$$w' = w - \eta \frac{\delta L}{\delta w},$$

where update term includes a factor of $\eta \cdot \gamma$. Since the optimal $\gamma$ was determined empirically between 0.01 and 2.0 as 0.01, the effective learning rate becomes $10^{-5}$. Consequently, we maintain a higher learning rate for NRFL to compensate. To simulate label noise, we randomly flip x% of the labels in the dataset and generate two instances each for 30% and 60% noisy datasets.

### A.2 NLP-BASED CLASSIFICATION TASK (NEWS20 DATASET)

For this set of experiments, we train LLaMA for 100 epochs using a learning rate of $10^{-5}$ and a batch size of 40, with $\gamma = 0.01$. Dimensionality reduction on the embeddings generated by the model is performed using UMAP, with the following parameters:

- n_components: 2

- n_neighbors: 15

- min_dist: 1.0

- metric: 'euclidean'

- random_state: 21

To simulate label noise, we randomly flip x% of the labels in the dataset and generate a single instance of a 20% noisy dataset.

### A.3 INFORMATION EXTRACTION TASK (JSON DATASET)

For these experiments, the models were trained for 100 epochs with a learning rate of $10^{-5}$ for all methods, except for NRFL and WRLL, which used a learning rate of $10^{-4}$ for the reasons described earlier. A batch size of 40 was used, and $\gamma$ for NRFL was set to 0.01.

To simulate label noise, 20% and 50% of the samples in the training set were corrupted by randomly flipping entity names within the target JSON objects. For each noise level, two instances were created, and results were averaged across both.

### A.4 TRANSLATION TASK (MALLS DATASET)

For this set of experiments, the models were trained for 100 epochs with a batch size of 40 and a learning rate of $10^{-5}$. Note that only the first 300 instances of the MALLS dataset were used.

To introduce noise, we follow the methodology of Yang et al. (2024) et al.. For each data instance where noise is applied, only one of the perturbations listed in Table 7 is selected. Similarly, two instances of each noise level were created, and the reported results correspond to the mean and standard deviation across both instances.

### A.5 REASONING AND QNA TASK

For both of these experiments, we use a learning rate of $10^{-5}$, train for 10 epochs, and set a batch size of 40. Noise is added to 20% and 50% of the datapoints in both datasets. For GSM8k, noise is applied only to the final answer, which is a number, by randomly flipping its digits. For Open-BookQA, the correct answer is randomly flipped to an incorrect one.

| Operation Type | Subtype | Original | Perturbed |
|---|---|---|---|
| Label Change | Change Predicate | $P(A) \wedge R(B)$ | $R(A) \wedge R(B)$ |
| | Change Term | $\forall x\, P(x) \wedge P(B)$ | $\forall y\, P(x) \wedge P(B)$ |
| | | $\forall x\, P(x) \wedge P(B)$ | $\forall x\, P(x) \wedge P(x)$ |
| | Change Operator | $\forall x\, P(x) \wedge P(B)$ | $\forall x\, P(x) \vee P(B)$ |
| Insert | Insert Term | $\forall x\, P(x) \wedge P(B)$ | $\forall x\, \exists y\, P(x) \wedge P(B)$ |
| | | $\forall x\, P(x) \wedge P(B)$ | $\forall x\, P(x) \wedge P(x,B)$ |
| | Insert Negation | $P(A) \wedge P(B) \wedge P(C)$ | $P(A) \wedge \neg(P(B) \wedge P(C))$ |
| | Insert Formula | $P(A) \wedge P(B)$ | $P(A) \wedge P(B) \rightarrow R(C)$ |
| Delete | Delete Term | $\forall x\, \forall y\, P(x) \wedge R(x,y)$ | $\forall y\, P(x) \wedge R(x,y)$ |
| | | $\forall x\, \forall y\, P(x) \wedge R(x,y)$ | $\forall x\, \forall y\, P(x) \wedge R(y)$ |
| | Delete Negation | $\neg(P(A) \wedge P(B))$ | $P(A) \wedge P(B)$ |
| | Delete Formula | $P(A) \wedge P(B) \wedge P(C)$ | $P(A) \wedge P(C)$ |

Table 7: The list of all atomic perturbations.

| Loss Type | UQ | | | UA | | |
|---|---|---|---|---|---|---|
| | 0% | 20% | 50% | 0% | 20% | 50% |
| CE | 19.18 | *24.49* | 15.45 | 17.12 | *21.69* | 14.93 |
| MAE | 22.13 | 14.61 | 18.64 | 14.04 | 16.53 | 16.76 |
| RLL | 37.26 | **26.21** | **22.25** | *30.61* | **22.93** | **21.67** |
| NFL | 20.17 | 7.58 | 5.31 | 14.25 | 5.36 | 6.23 |
| NRFL | **41.56** | 16.32 | *19.62* | **47.73** | 20.45 | *18.93* |
| WRLL | 9.42 | 8.97 | 9.5 | 6.25 | 5.67 | 5.69 |

Table 8: Accuracy of LLaMa trained on ASAG dataset

# B    EXTENDED EXPERIMENTAL EVALUATION

## B.1    AUTOMATIC SHORT ANSWER GRADING TASK

Finally, we consider an Automatic Short Answer Grading task, given a question and a reference answer, and the LLM must generate a score for a student's response. This task requires both natural language understanding and comparative reasoning capabilities between the two answers. Results indicate that NRFL outperforms other methods in most cases. In contrast, WRLL performs poorly, as it is better suited for classification tasks with label noise. Given the reasoning-intensive nature of this task, NRFL is more effective than WRLL. We conjecture that noise-robust reinforcement learning techniques may be better suited for such tasks than standard supervised fine-tuning using noise robust loss functions.

## B.2    EVALUATION OF RESNET18 ON CIFAR-100 DATASET

| Loss Function | 0% | 30% | 60% |
|---|---|---|---|
| CE | 52.87 | 51.43 | 41.60 |
| RLL | 41.53 | 41.02 | 26.44 |
| NFL | 12.43 | 9.01 | 6.10 |
| GCE | 51.92 | 53.50 | 39.38 |
| NRFL | **55.98** | 49.01 | 35.29 |
| WRLL | 54.45 | **52.29** | **42.09** |

Table 9: Accuracy of ResNet-18 after training on CIFAR-100 dataset

ResNet-18 has been trained on the CIFAR-100 dataset for 400 epochs, along with a learning rate of 0.1 for all loss functions. It can again be observed from Table 9 that our methods outperform others, including recent loss functions like GCE. We attribute this improvement to the model's ability to learn more discriminative decision boundaries, consistent with the patterns observed in earlier experiments (as in Figure 1).

