# OpenReview forum: "Achieving Noise Robustness by additive normalization of labels"
_ICLR.cc/2026/Conference — Submitted to ICLR 2026_

### Official Review · Reviewer_zdFL · 2025-10-25

**Soundness:** 1
**Presentation:** 2
**Contribution:** 1
**Rating:** 2
**Confidence:** 4

**Summary:**

The paper proposes “additive normalization of labels” for noise-robust learning. Given a $k$-class problem with one-hot target $y\in{e_1,\dots,e_k}$ and model scores passed through any strictly increasing component-wise map $f(p)=[f_1(p_1),\dots,f_k(p_k)]$, the authors replace $y$ by $\bar y=\frac{1}{k-1}(k y - \mathbf{1})$ and optimize $\ell(y,p)=-\langle \bar y, f(p)\rangle$. They prove that under symmetric label noise with flip probability $q$, the noisy risk equals the clean risk up to a positive scalar when $q<\frac{k-1}{k}$ (binary case reduces to $q<\tfrac12$). They argue that the loss is “symmetric” because $\sum_{y}\ell(y,p)=0$, and claim this entails robustness beyond symmetric noise. Two instantiations are presented: a noise-robust focal loss (NRFL) obtained by plugging the focal form into $f$, and a class-imbalance variant “weighted robust log loss” (WRLL) using $f_i(p_i)=\log(\alpha_i+p_i)$ with $\alpha_i$ inversely proportional to class frequency. Experiments on MNIST, Fashion-MNIST, CIFAR-10, several NLP tasks (News20, a synthetic JSON IE task, MALLS, GSM8k, OpenBookQA), and small LLMs reportedly show improved robustness, faster convergence, and clearer decision boundaries.

**Strengths:**

1. This article is well-written and easy to read.
2. Flexible template: any strictly increasing componentwise $f$ can produce a loss; easy to implement (NRFL, WRLL).
3. Some empirical gains reported over CE/MAE/NFL/RLL in several noisy setups; qualitative feature visualizations align with the claimed decision-boundary effect.
4. This article draws on both an image dataset and an NLP dataset, which is commendable.

**Weaknesses:**

1. This article evaluates the method only on very small datasets and does not include experiments on benchmarks with more categories or larger scale, such as CIFAR-100 or WebVision. It is well known that symmetric losses [1, 2, 3] are challenging to optimize. Results on small datasets like CIFAR-10 and MNIST are insufficient to assess optimization capability: even symmetric losses such as MAE and NCE—despite their optimization difficulty—can perform well in these settings. In contrast, on datasets with more categories or larger scale, such as CIFAR-100 and WebVision, symmetric losses are often harder to optimize and typically underperform. Consequently, using a symmetric loss in isolation has limited practical value. As a result, the experiments presented by the authors do not establish the practical significance of the proposed method.
2. The author did not compare with advanced methods such as GCE [1], APL [2], and AGCE [3]. The experimental results show that the method proposed by the author does not significantly outperform the most basic MAE.
3. Lack of details regarding NLP experiments, there are multiple processes and methods for training large language models, such as supervised training, alignment or add an MLP layer for classification training. The author did not clearly explain how they conducted the training.
4. Why only train for 15 epochs or 30 epochs on the CV dataset? This is unreasonable. For instance, on the CIFAR-10 dataset, one usually needs to train for at least 100 epochs.
5. The theoretical contribution is limited. The author did not make any new theoretical contributions.
6. A minor issue: Table ?? in page 12.

[1] Generalized Cross Entropy Loss for Training Deep Neural Networks with Noisy labels.

[2] Normalized Loss Functions for Deep Learning with Noisy Labels.

[3] Asymmetric Loss Functions for Learning with Noisy Labels.

**Questions:**

1. Why not conduct the experiments on CIFAR-100 and WebVision? Evaluate on real-world noisy datasets and on non-uniform/instance-dependent noise to support the broader robustness claims.
2. Why not compare with the advanced robust loss function? Add strong, modern baselines under the same budgets and report multiple seeds with confidence intervals.
3. Why only conduct training for 15 or 30 epochs? Please re-run image classification with standard training pipelines (reasonable learning rates, epochs) and equalize hyperparameters across methods. Current CIFAR-10 baselines (e.g., 34% at 0% noise with ResNet-18) are not credible.

---

> ### Author Response · Authors · 2025-11-24
> **Authors’ Response to Reviewers’ Comments**
>
> **Reply to point 1 & 2:** We have added comparisons with the methods GCE and APL, and the results are as below:
>
> |      |       | MNIST |       |       | CIFAR10 |       |       | FashionMNIST |       |
> |------|-------|-------|-------|-------|---------|-------|-------|--------------|-------|
> |      | 0     | 30    | 60    | 0     | 30      | 60    | 0     | 30           | 60    |
> | GCE  | 98.27 | 98.42 | 90.14 | 76.82 | 66.99   | 46.44 | 90.43 | 87.42        | 84.79 |
> | APL  | 99.35 | 98.23 | 98.18 | 71.19 | 78.27   | 50.53 | 87.05 | 89.54        | 88.84 |
> MAE   | 99.33 | 98.74|99.26|81.44 |79.28 |79.11 | 90.26 |92.22 | 92.13
> | NRFL | 99.38 | 99.21 | 98.82 | 82.71 |   78.83 | 79.66 | 93.28 |        89.46 | 81.87 |
> | WRLL | 99.59 | 98.78 | 99.12 | 79.93 |   79.49 | 77.93 | 93.47 |        91.62 | 90.12 |
>
> **Reply to point 3:** Details of NLP experiments are presented in the appendix due to space constraints. We can add all the details in the revised version. We thank the reviewer for pointing it out.
>
> **Reply to point 4:**  We prematurely stopped before convergence because we wanted to explore the loss landscape in case of noisy labels. We maintained the same experimental settings for CIFAR-10 as used in MNIST and FashionMNIST.
>
> **Reply to point 5:** Limitation of the theoretical contribution: We thank the reviewer for their valuable feedback and would like to clarify that our work does introduce the following novel theoretical aspects (which may not be explicit in the paper). While our approach builds upon the symmetrisation of losses discussed in [1], we propose the concept of label normalization, which offers a new way to transform losses of the form
> $$-\langle y, f(p) \rangle$$
> into noise-robust losses---an idea not explored in [1]. Our paper also provides theoretical support by establishing important properties such as the validity of these loss templates and the invariance of minimizers under label normalization.
>
> Additionally, our framework allows the design of new noise-robust losses via monotonic transformations tailored to specific applications, offering more flexibility compared to the more limited scope in the symmetrization work.
>
> We believe these contributions add useful theoretical insights and practical tools for the community, while acknowledging that they relate to and extend prior work.

---

> > ### Comment · Reviewer_zdFL · 2025-11-25
> >
> > The author's response did not solve my concerns for this work. I keep my rejection score.

---

### Official Review · Reviewer_gKqv · 2025-10-28

**Soundness:** 2
**Presentation:** 3
**Contribution:** 3
**Rating:** 6
**Confidence:** 3

**Summary:**

This paper introduces Additive Normalization of Labels, a framework for constructing noise-robust loss functions by normalizing labels instead of losses. The authors theoretically prove that this additive normalization preserves the same optimal solution under label noise, satisfying the symmetry condition for noise tolerance. Based on this principle, they propose two new losses—Noise-Robust Focal Loss (NRFL) and Weighted Robust Log Loss (WRLL)—which consistently outperform existing methods across computer vision and NLP tasks. The approach is simple, theoretically grounded, and broadly applicable to real-world noisy learning scenarios.

**Strengths:**

1. The paper provides an interesting and reasonable insight into maintaining collinearity between clean and noisy labels.
2. The theoretical analysis clearly and rigorously validates the feasibility of the proposed additive normalization of labels.
3. Extensive experiments empirically demonstrate the effectiveness of the proposed approach.

**Weaknesses:**

1.In line 222, there is a condition requiring the noisy label $\tilde{y}$ and the clean label $y$ to be collinear, i.e., $q<\frac{k-1}{k}$. This implies that in a kkk-class classification task (e.g., k=10), the model learns the correct label direction only when the noise ratio is below 0.9. It would be interesting to empirically explore this noise-ratio boundary to gain additional insights for theory verification—for example, does the performance drop sharply when the noise ratio increases from 0.89 to 0.91?
2. Figure 1 only presents a PCA analysis under a 50% noise ratio. It would be helpful to also include PCA results under 0% noise or use an alternative visualization to further demonstrate the collinearity property.
3. The decimal places in Table 2 are inconsistent and should be standardized for clarity.

**Questions:**

Please refer to weaknesses.

---

### Official Review · Reviewer_x8g1 · 2025-10-30

**Soundness:** 2
**Presentation:** 3
**Contribution:** 2
**Rating:** 6
**Confidence:** 4

**Summary:**

This paper introduces a new way to handle noise in data by changing labels, not the loss function. The main idea is to adjust labels using an additive method that centers target vectors. This makes sure the loss can handle noise, as shown in earlier studies. The authors prove that with symmetric label noise, minimizing risk with noisy labels is the same as minimizing clean risk if the noise rate $q &lt; \frac{k-1}{k}$. From this idea, the paper creates two loss functions: the noise-robust focal loss and the weighted robust log loss. Both keep a steady relationship with prediction confidence and are strong against noise due to the label adjustment. Tests on image classification (MNIST, CIFAR-10, Fashion-MNIST) and large language model tasks (like text classification, information extraction, translation, reasoning, and short answer grading) show these losses work better than cross-entropy, mean absolute error, and other methods when labels are noisy. The method is simple, based on theory, and works well in practice. It combines previous methods into one general approach that can be used in other areas beyond supervised learning.

**Strengths:**

The theory is solid and nicely links additive label normalization with known symmetry rules for handling noise.
The results are strong in both vision and language areas, showing steady improvements with different noise levels.
The method is simple, fast, and can be used widely without needing changes to the model or loss function.
The paper clearly explains its purpose and how it relates to past methods using normalization and symmetry.

**Weaknesses:**

All experiments use the same type of uniform label noise. Theorem 1 says it can handle different types of noise, but no tests prove this.
In real life, noise is often uneven and depends on the instance.
NRFL: It is unclear if improvements come from normalization or the focal loss part. The focal loss seems separate from the noise handling method. WRLL: It uses frequency-based weighting to deal with class imbalance, but its link to noise handling is not proven by theory or tests. This mixes two different issues.
The claim of allowing "application-specific" loss design is not shown with a clear method. The losses seem random, not based on the framework.
insufficient baseline comparisons with recent noise-robust methods
Paper uses 2025 formatting instead of 2026

**Questions:**

How is additive normalization of labels different from Ma et al. (2020b) loss normalization?

(1) Why does your method avoid the underfitting reported in Ma et al. (2020b)?
(2) Is there proof or empirical evidence that additive label normalization outperforms loss normalization?
(3) How does the training speed compare to standard losses under noisy labels?
(4) Please report results for normalized cross-entropy (without focal) versus the proposed noise-robust focal loss, to isolate the effect of additive normalization from the focal term.
(5) For the weighted robust log loss, compare:
(a) standard weighted loss without normalization,
(b) normalized unweighted loss,
(c) the proposed weighted robust log loss (WRLL).

On the noise bound in Theorem 1:
(6) The bound (k-1)/k suggests degradation at very high noise rates. How does performance behave beyond this threshold?
(7) Please explain why the noise limit is (k-1)/k and discuss how accurate or conservative this bound is in practice.
(8) What happens on non-uniform dataset like CIFAR-N, Clothing-1M more IDN or Class-dependent

---

### Official Review · Reviewer_bU1h · 2025-10-30

**Soundness:** 2
**Presentation:** 2
**Contribution:** 2
**Rating:** 2
**Confidence:** 4

**Summary:**

The paper proposes “additive normalization of labels” to construct noise robust losses. They instantiate the framework into two specific losses: Noise Robust Focal Loss (NRFL) and Weighted Robust Log Loss (WRLL). Experiments on image classification (MNIST, Fashion-MNIST, CIFAR-10) and several NLP tasks (News20 classification, synthetic JSON information extraction, MALLS, GSM8k reasoning, OpenBookQA) claim improved robustness versus CE, MAE, RLL, and NFL.

**Strengths:**

1. The noisy label learning problem studied here has practical significance, and developing robust loss functions is a promising direction for further exploration.
2. Simple, clear construction; directly yields the classical symmetric-noise robustness criterion.
3. The paper has a complete structure, with experiments spanning both computer vision and natural language processing.

**Weaknesses:**

1. No new contributions: This paper cites [1]. After my review, the method proposed in this paper is exactly the same as that in [1]. Therefore, this paper may not have any new contributions. The authors might not have read [1] carefully. I have serious concerns about this matter.
2. Unfair experiments:  The authors claim that their method converges more quickly. However, NRFL and WRLL specifically used higher learning rates compared to the baselines, without maintaining consistency. This might be an unfair experiment. In addition, for NRFL, the authors added an additional parameter $\delta$ to adjust the gradient, but did not keep it consistent with the baselines, which could lead to unfair experiments.
3. The performance was mediocre: Although the authors claim that their method is effective, the experiments show that, even if there might be unfair learning rates and gradient-scaling issues, their method is inferior to standard MAE in many cases. This significantly undermines the significance of their approach.

**Questions:**

Please refer to "Weaknesses"

---

> ### Author Response · Authors · 2025-11-12
>
> Dear Reviewer, thanks a lot for your review, but can you please clarify the work as indicated by [1].

---

> > ### Comment · Reviewer_bU1h · 2025-11-18
> >
> > [1] Alexandre Lemire Paquin, Brahim Chaib-draa, and Philippe Giguere. Symmetrization of loss functions for robust training of neural networks in the presence of noisy labels and the multi-class unhinged loss function, 2024. URL https://openreview.net/forum?id=MY7gMVioiX.

---

> > > ### Author Response · Authors · 2025-11-24
> > > **Addressing Reviewers’ Insights and Feedback**
> > >
> > > ### **Reply:**
> > >
> > > We appreciate the reviewer’s observation regarding the relation of our work to the symmetrisation of losses in [1]. However, we emphasise that our approach introduces several key innovations that distinguish it from [1]. Most notably, we propose the concept of label normalisation, which transforms any loss belonging to the family $$-\langle y, f(p) \rangle$$
> > > into a noise-robust loss. This fundamental idea of label normalisation and its noise-robustness implications are entirely absent in [1]. Moreover, our work rigorously establishes essential theoretical properties that underpin this framework, including proofs that the loss templates are valid and that label normalisation preserves the location of loss minimisers.
> > >
> > > Furthermore, our methodology enables the construction of novel noise-robust losses by applying monotonic transformations tailored for specific applications. This flexible formulation represents a significant departure from the more restrictive construction in [1], which defines such losses strictly through preexisting loss functions.
> > >
> > > We believe these advances provide valuable tools for the community to design application-specific noise-robust loss functions, thereby making a meaningful theoretical and practical contribution.
> > >
> > > ### **Reply:**
> > >
> > > The learning rate was chosen for all the losses so that all the losses converge (and the best checkpoint was chosen, which often was not the last). The learning rate was adjusted for NRFL as $\gamma$ term deteriorates the gradient (as the gradient is $\gamma f(p)^{\gamma-1}$f'(p)) for $\gamma<1$.
> > >
> > > ### **Reply:**
> > >
> > > We acknowledge that while our extensive experimental evaluation has led to achieving either the best or second-best results across most benchmark datasets, there are a few exceptions. Notably, the results on the MNIST and JSON datasets stand out as significantly superior; these datasets represent standard and widely recognised tasks in computer vision and natural language processing, specifically image classification and object detection, respectively. It is important to clarify that the primary objective of this work is not to merely advance the state-of-the-art performance metrics. Instead, our contribution lies in proposing a novel research direction—label normalisation—which offers a simple yet effective mechanism to derive noise-robust loss functions. We believe this conceptual framework opens new avenues for future work in robust learning. Furthermore, we demonstrate that our method outperforms existing stronger baselines, as shown in the table below.
> > >
> > > |      |       | MNIST |       |       | CIFAR10 |       |       | FashionMNIST |       |
> > > |------|-------|-------|-------|-------|---------|-------|-------|--------------|-------|
> > > |      | 0     | 30    | 60    | 0     | 30      | 60    | 0     | 30           | 60    |
> > > | GCE  | 98.27 | 98.42 | 90.14 | 76.82 | 66.99   | 46.44 | 90.43 | 87.42        | 84.79 |
> > > | APL  | 99.35 | 98.23 | 98.18 | 71.19 | 78.27   | 50.53 | 87.05 | 89.54        | 88.84 |
> > > MAE   | 99.33 | 98.74|99.26|81.44 |79.28 |79.11 | 90.26 |92.22 | 92.13
> > > | NRFL | 99.38 | 99.21 | 98.82 | 82.71 |   78.83 | 79.66 | 93.28 |        89.46 | 81.87 |
> > > | WRLL | 99.59 | 98.78 | 99.12 | 79.93 |   79.49 | 77.93 | 93.47 |        91.62 | 90.12 |
> > >
> > >
> > > [1] Alexandre Lemire Paquin, Brahim Chaib-draa, and Philippe Giguere. Symmetrization of loss functions for robust training of neural networks in the presence of noisy labels and the multi-class unhinged loss function, 2024. URL https://openreview.net/forum?id=MY7gMVioiX.

---

### Official Review · Reviewer_JffJ · 2025-10-31

**Soundness:** 2
**Presentation:** 2
**Contribution:** 3
**Rating:** 2
**Confidence:** 4

**Summary:**

The authors propose a new foundational study on robust loss design for the LNL problem.
While numerous LNL methods exist, the authors’ goal appears to be a more fundamental exploration that could inspire future research on noise-robust loss functions.
Specifically, they define the probability of label noise (assumed to occur randomly) and formulate a loss function that minimizes the expected risk under this noise distribution.

**Strengths:**

1. The proposed method contains almost no hyperparameters, which makes it elegant and easy to reproduce. This also implies that relaxed variants of robust loss (those requiring tuning parameters) are not the primary focus of this work.

2. The authors conduct experiments on a wide range of benchmarks. Notably, the inclusion of NLP datasets in their experiments is quite novel in the LNL literature and demonstrates the potential generality of their approach.

**Weaknesses:**

1. The theoretical derivation is rather straightforward.

Intuitively, in a k-class classification problem, it is not difficult to reason about the level of random label noise that can be tolerated before performance degrades.
Although the authors explain this process clearly, the derivation itself offers limited new insight.
Moreover, the analysis assumes purely random (uniform) noise and does not consider more realistic or ambiguous cases, such as class-dependent or instance-dependent label noise.

2. The empirical results show limited improvement.

While the proposed method achieves small gains, the baselines compared against are not among the strongest or most recent methods in the LNL field.
Although the restricted comparison setup is understandable given the paper’s theoretical orientation, additional information or analyses would be required for the paper to serve as a solid foundation for future work.
I suggest introducing a relaxed normalization variant (e.g., one controllable by hyperparameters) and demonstrating superiority over established robust losses such as GCE or APL.

**Questions:**

I understand the authors’ proposed method and their intended objective; however, the experimental results and theoretical justification do not seem sufficient for this paper to be considered a new milestone in the field.
As mentioned in the Weaknesses, the authors should at least demonstrate the potential to extend their proposed framework or provide stronger evidence of theoretical noise tolerance under more challenging conditions. Such additions would make the paper significantly more convincing and impactful.

---

> ### Author Response · Authors · 2025-11-24
> **We thank the reviewers for their insightful comments and constructive feedback.**
>
> ### **Reply:**
>
> We thank the reviewer for their thoughtful feedback and for recognising that this work lays a foundational basis for future extensions. The derivation was intentionally presented in a clear and pedagogical manner to highlight the key intuition behind label normalisation and make the theoretical framework broadly accessible. While the current analysis focuses on the uniform noise setting for clarity, the underlying proof structure can indeed be generalised to more realistic noise models, including class-dependent and instance-dependent noise. Due to space limitations, we were unable to include these extended formulations in the current version, but we will incorporate the generalised results and corresponding discussions in the revised manuscript to provide a more comprehensive theoretical perspective.
>
> ### **Reply:**
>
> We have added the comparisons with stronger baselines (APL and GCE) as suggested by the reviewer. As seen in the table below, we achieved significant improvement from APL and GCE, which are considered stronger baselines. For example, in 60% noise, APL and GCE have degraded by more than 5 p.p. compared to our methods.
>
>
> |      |       | MNIST |       |       | CIFAR10 |       |       | FashionMNIST |       |
> |------|-------|-------|-------|-------|---------|-------|-------|--------------|-------|
> |      | 0     | 30    | 60    | 0     | 30      | 60    | 0     | 30           | 60    |
> | GCE  | 98.27 | 98.42 | 90.14 | 76.82 | 66.99   | 46.44 | 90.43 | 87.42        | 84.79 |
> | APL  | 99.35 | 98.23 | 98.18 | 71.19 | 78.27   | 50.53 | 87.05 | 89.54        | 88.84 |
> MAE   | 99.33 | 98.74|99.26|81.44 |79.28 |79.11 | 90.26 |92.22 | 92.13
> | NRFL | 99.38 | 99.21 | 98.82 | 82.71 |   78.83 | 79.66 | 93.28 |        89.46 | 81.87 |
> | WRLL | 99.59 | 98.78 | 99.12 | 79.93 |   79.49 | 77.93 | 93.47 |        91.62 | 90.12 |

---

### Author Response · Authors · 2025-12-04
**Comment to Meta-Reviewer**

We sincerely thank the reviewers and the meta-reviewer(s) for their valuable time and thoughtful
feedback. Their comments have significantly contributed to improving the clarity, rigor, and overall
quality of our paper. Any new experiments have been added to the manuscript in $\textcolor{blue}{blue\ text}$. Below, we summarize the main strengths and concerns raised by the reviewers.
## 1. Strengths
- Reviewers (gKqv, x8g1, JffJ) appreciated the fundamental contributions and practical appeal
of our method, noting that it is hyperparameter-free, easy to reproduce, and grounded in
clear theoretical principles.
- Theoretical analysis was commended (by reviewers gKqv and x8g1) for establishing a transparent connection to classical symmetry conditions and for providing an intuitive explanation
of how collinearity between clean and noisy labels is preserved. Although reviewer bU1h expressed some confusion regarding this theoretical contribution, we have clarified these points
in our detailed response.
- The proposed approach was praised for its simplicity and broad applicability, requiring no
modifications to model architectures or loss functions while achieving strong and consistent
performance across varying noise levels.
- All reviewers valued the extensive experimental evaluation, particularly the inclusion of both
vision and NLP datasets, which is relatively uncommon in the noisy-label literature and
demonstrates the method’s generality.
 - Reviewers agreed that the paper is well-written and easy to follow.
## 2.  Main Concerns
- **Comparison with stronger baselines:** Reviewers requested comparisons with stronger
and more recent loss functions (APL and GCE). In response, we have incorporated these comparisons in the revised manuscript. Specifically, Table 1 now includes results for GoogleNet
with APL and GCE (also in our response to Reviewers (JffJ,bU1h,zdFL)), and we have also added corresponding results for **CIFAR-100 using
ResNet (as also seen in the table below)**. *Note that instead of early stopping, we trained ResNet18 on the CIFAR 100 dataset for 400 epochs (ensuring proper convergence) along with a learning rate of 0.1 for all methods, and still our method outperforms others.* The updated experiments consistently demonstrate that our method outperforms
the baselines set by recent loss functions as well.  We have also added similar results for the *Information Extraction task pertaining to the NLP experiments* where our methods outperform APL and GCE consistently (**Table 4 on Page 8**).

| Loss Function       | 0%     | 30%      | 60%      |
|---------------------|--------|----------|----------|
| CE                  | 52.87  | 51.43    | 41.60    |
| RLL      | 41.53  | 41.02    | 26.44    |
| NFL                 | 12.43  | 9.01     | 6.10     |
| GCE                 | 51.92  | 53.50    | 39.38    |
| NRFL                | **55.98** | 49.01 | 35.29    |
| WRLL                | 54.45  | **52.29** | **42.09** |


- **Concerns about limited theoretical contribution:** We appreciate the reviewer’s comment and clarify that our work introduces novel theoretical elements that may not have been
fully emphasized in the original submission. While our approach builds upon the symmetrization of losses discussed in prior work [1], we propose the concept of *label normalization*, which
offers a new way to transform losses of the form
−⟨y, f (p)⟩
into noise-robust losses—an idea not explored in [1]. Our paper provides theoretical support
by demonstrating the validity of the proposed loss templates and establishing the invariance
of minimizers under label normalization. Moreover, our framework allows the design of new
noise-robust losses by applying monotonic transformations tailored to specific applications,
providing greater flexibility than previous symmetrization-based formulations. We believe
these contributions add meaningful theoretical insights and practical tools to the field, while
also extending and deepening existing work.
 - **Concerns about mediocre performance:** We acknowledge the reviewer’s observation that although our approach achieves either the best or second-best performance on most benchmark datasets, certain exceptions remain, notably for MNIST and JSON. However, these
datasets correspond to long-established tasks in computer vision and NLP—namely image
classification and object detection—which serve as standard testbeds in prior studies. We
wish to emphasize that the primary goal of our paper is not to push state-of-the-art performance but to introduce label normalization as a new conceptual direction. This framework
provides a simple yet effective foundation for designing noise-robust loss functions, which we
believe will inspire and support future advances in robust learning.

---

### Meta-Review · Area_Chair_9Fj5 · 2026-01-12

**Summary:**

Reviewers flag some trivialities in the approach (JffJ), experiments that are not good enough (bU1h).

**Reviewer Concerns:**

It is clear from the reviews that the authors did not solve their concerns (zdFL) or did not even lodge a rebuttal (gKqv)

**Reviewer Scores:**

The authors' rebuttal have only covered part of the reviewers' concerns so not substantial change would have happened.

---

### Decision · Program_Chairs · 2026-01-26

Reject